# Poultry Slaughterhouse Wastewater Treatment Using an Integrated Biological and Electrocoagulation Treatment System: Process Optimisation Using Response Surface Methodology

Philadelphia Vutivi Ngobeni [1], Larryngeai Gutu [1], Moses Basitere [2,*], Theo Harding [1] and David Ikumi [1]

1   Water Research Group, Department of Civil Engineering, University of Cape Town, Rondebosch, Cape Town 7700, South Africa; phila.ngobeni@uct.ac.za (P.V.N.); larrahgutu@icloud.com (L.G.); theo.harding@uct.ac.za (T.H.); david.ikumi@uct.ac.za (D.I.)
2   Academic Support Programme for Engineering in Cape Town (ASPECT), Centre for Higher Education Development, University of Cape Town, Rondebosch, Cape Town 7700, South Africa
*   Correspondence: moses.basitere@uct.ac.za

**Abstract:** The feasibility of a biological (Ecoflush[TM]) and/or electrocoagulation (EC) treatment system in removing chemical oxygen demand (COD) and fats, oils, and grease (FOG) from poultry slaughterhouse wastewater (PSW) were studied. The response surface methodology (RSM) was used to identify the optimum operating condition for EC and its integration with Ecoflush[TM] as a pre-treatment for the removal of lipids. The optimum operating conditions were obtained at a pH of 3.05, a current density of 66.9 A/m$^2$, 74-min of treatment time, and without Ecoflush™. These conditions produced a high-quality clarified effluent after 92.4% COD reduction and 99% FOG reduction. The treatment with Ecoflush[TM] only resulted in 85–99% FOG reduction, 20–50% COD reduction, and odourless effluent. However, the combination of both processes (Ecoflush[TM] and EC) did not yield a significant difference (F test, $p > 0.05$) when compared to the performance of EC alone. Despite the low removal percentages of nitrogen and phosphorus, the present study proved that EC is an effective method for the removal of COD and FOG, rendering an effluent that meets the permissible discharge standards for the City of Cape Town. The novel Ecoflush™ also proved to be very efficient in the removal of FOG from PSW.

**Keywords:** poultry slaughterhouse wastewater; EcoFlush[TM]; electrocoagulation; response surface methodology

## 1. Introduction

The poultry industry holds a prominent position among livestock-based trades due to its enormous potential to drive rapid economic growth [1]. Although this industry has notable achievements in socio-economic development, the volume of effluent generated is also very high [2]. Given global and local challenges such as water scarcity, the pollution of surface water, the spread of water-borne diseases, and penalties imposed by regulatory authorities on industries for the discharge of untreated wastewater, it is critical for industries to select and implement enhanced wastewater-treatment strategies aimed at reducing the concentration of contaminants [3]. As alternatives to the activated sludge process, wastewater-treatment strategies such as electrooxidation, electroflotation, and electrocoagulation have been investigated in the last decade specifically in the treatment of industrial wastewater streams [4]. There is an increased focus on investigating the application of electrocoagulation in the treatment of poultry slaughterhouse wastewater (PSW).

Electrocoagulation (EC) is an electrochemical process that presents an excellent alternative for poultry slaughterhouse wastewater (PSW) treatment, as this treatment method can handle fluctuations in pollutant quality and quantity and can remove persistent pollutants

from wastewater [5,6]. The main advantages of the EC process are (i) rapid breakdown of organic compounds, (ii) no addition of supplementary compounds, (iii) environmental compatibility, (iv) high efficiency in pollutant degradation, and (v) cost-effectiveness. EC is based on passing electric current to degrade organic contaminants via redox reactions [7]. Through oxidation (Equation (1)), the anode generates metal cations, hydroxyl ions ($OH^-$), and dihydrogen molecules ($H_2$) by water reduction at the cathode (Equation (2)). In Equations (3)–(5), $OH^-$ reacts with metal cations in the reaction medium to form metal hydroxides. The latter plays a significant role in pollutant removal in the EC process by (i) adsorption, (ii) coagulation, and (iii) flotation. The electrodes can be arranged in a monopolar or bipolar mode. The most common materials of construction for electrodes are aluminium and iron in plate forms [8]. The following are simplified reactions occurring at the electrodes (iron):

$$\text{Oxidation (cathode)}: \ Fe \rightarrow Fe^{2+} + 2e^- \tag{1}$$

$$\text{Reduction (anode)}: \ 2H_2O + 2e^- \rightarrow H_2 + 2OH^- \tag{2}$$

$$\text{Reaction medium}: \ O_2 + Fe^{2+} + 2H_2O \rightarrow 4Fe^{3+} + 4OH^- \tag{3}$$

$$Fe^{2+} + 2OH^- \rightarrow Fe(OH)_2 \tag{4}$$

$$Fe^{3+} + 3OH^- \rightarrow Fe(OH)_3 \tag{5}$$

The efficient treatment of lipid-rich wastewater, such as PSW, which contains more than 67% of the wastewater's particulate chemical oxygen demand (COD) using EC, poses considerable technical and economic challenges as lipids generally inhibit the process [9]. Operational challenges such as electrode passivation, which occurs when an impermeable film is deposited on the surface of electrodes, reduce current, and, as a result, the intensity of the redox process efficiency, is one of the major problems for EC [10]. The contents in the lipids may be challenging to degrade using EC, as the fats, oil, and grease (FOG) may coat the electrode, thus creating a barrier for electrical conduction [11]. Furthermore, this wastewater contains significant phosphorus, nitrogen, organic carbon, heavy metals, and nutrients, attributed to residual blood, excreta, detergents, disinfectants, and other chemicals used in the process [1]. EC is also ineffective at eliminating ammonia-bound nitrogen.

In this context, biological systems under aerobic conditions using EcoFlush^TM (a novel bacteria-enzymatic consortium blend that is used for the remediation of hydrocarbons) can achieve high organic matter and nutrient removal efficiencies. This technology has already showcased its efficacy in remediating PSW [12,13]. The studies reported removal efficiencies of 38–56% TSS, 50–70% COD, and 80–82% FOG. Therefore, it is hypothesized that EC combined with a biological system under aerobic conditions using EcoFlush^TM can be an effective method for a highly efficient operation to achieve a satisfactory quality for the final treated effluent, especially for this type of wastewater that contains various contaminants.

Therefore, the purpose of this study was to investigate the significance of a biological treatment under aerobic conditions combined with electrocoagulation to treat wastewater generated from the poultry industry. The work involved the characterisation of the wastewater, assessment of biodegradation under aerobic pre-treatment conditions using Ecoflush^TM, and optimization of electrocoagulation on COD and FOG reduction under various operating parameters, including initial pH, current density, and reaction time. To date, no research has been conducted on the removal efficiencies of the aerobic/electrocoagulation process using response surface methodology (RSM) in PSW. Furthermore, electrocoagulation systems are still being improved as a long-term sustainable technology.

## 2. Materials and Methods

### 2.1. Poultry Slaughterhouse Wastewater Source

The poultry slaughterhouse wastewater (PSW) used in this study was collected from a poultry slaughterhouse located in the Western Cape, South Africa (SA). The facility slaugh-

ters 20,000 chickens per day, producing approximately 450 m$^3$ of PSW daily. The wastewater emerging from various operations such as stunning and slaughtering, de-feathering, evisceration, trimming, carcass washing, de-boning, chilling, packaging, cleaning of facilities and equipment was filtered using a 10–30 mm mechanical screen to remove suspended solids before further treatment. The PSW samples used in this study were obtained by grab sampling with plastic scoops from an equalisation tank, collected in separate polypropylene airtight storage containers, and kept at 4 °C in a refrigerator in the laboratory before use. Aliquots were then sampled for PSW characterisation.

### 2.2. Characterisation of Poultry Slaughterhouse Wastewater

The PSW samples were characterised for chemical oxygen demand (COD), fats, oils, and grease (FOG), suspended solids (SS), ammonia nitrogen (NH$_3$-N), phosphates (PO$_4^{3-}$), heterotrophic plate count, total coliform, and *Escherichia coli* (E-coli), which were analysed at Bemlab (Somerset West, SA) using standard methods from the Environmental Protection Agency (EPA) for water and wastewater analyses shown in Table 1. Physicochemical parameters such as pH, salinity, conductivity, and total dissolved solids (TDS) were measured using a multi-parameter instrument (Senso-Direct 150, Springfield, IL, USA), and turbidity was determined using a turbidimeter (HI-93414-02, HANNA, Smithfield, CA, USA).

**Table 1.** Methods used for characterizing poultry slaughterhouse wastewater [14].

| Parameters | Method |
|---|---|
| COD | EPA method 3289 |
| FOG | EPA method 10,056 |
| SS | EPA method 4993 |
| NH$_3$-N | EPA method 4511 |
| PO$_4^3$-P | EPA method 4511 |
| Heterotrophic Plate Count | EPA method 1454 |
| Total coliforms | EPA method 6386 |
| *E. Coli* | EPA method 6386 |

### 2.3. Experimental Setup and Procedure Wastewater

The experimental setup used in this study is shown in Figure 1.

#### 2.3.1. Biological Pre-Treatment Process

The biological pre-treatment of PSW was carried out in a 25 L polypropylene container at ambient temperature (24 °C). A volume of 100 mL Eco-flush™ (Mavu Biotechnologies Pty Ltd., Cape Town, South Africa) was added to 20 L of raw PSW. The mixture was aerated for 24 h using a Resun Ac 9906 six-outlet air pump (Hydroponic, Port Elizabeth, South Africa) to sparge air into the pre-treatment tank using silicone tubes connected to two diffusers that provided sufficient micro-bubble formation into the system. The micro-bubble formation ensured a steady, adequate supply of dissolved air for optimal aerobic bacteria proliferation. The aerated mixture settled for a further 24 h. This allowed sufficient time for the Eco-Flush™ to digest the FOG and protein found in the PSW. Skimming was carried out to remove the FOG. In the process of scrapping, some solids trapped in the FOG were removed. After settling, the mixture was strained using two sieves with apertures of 1.18 mm and 53 μm, respectively. The strained product was recycled into a 25 L holding tank. This product was fed into the electrocoagulation (EC) reactor as the raw feed. The operating conditions used in this study were reported in a previous study by [15], who optimised the system while treating PSW from the same site.

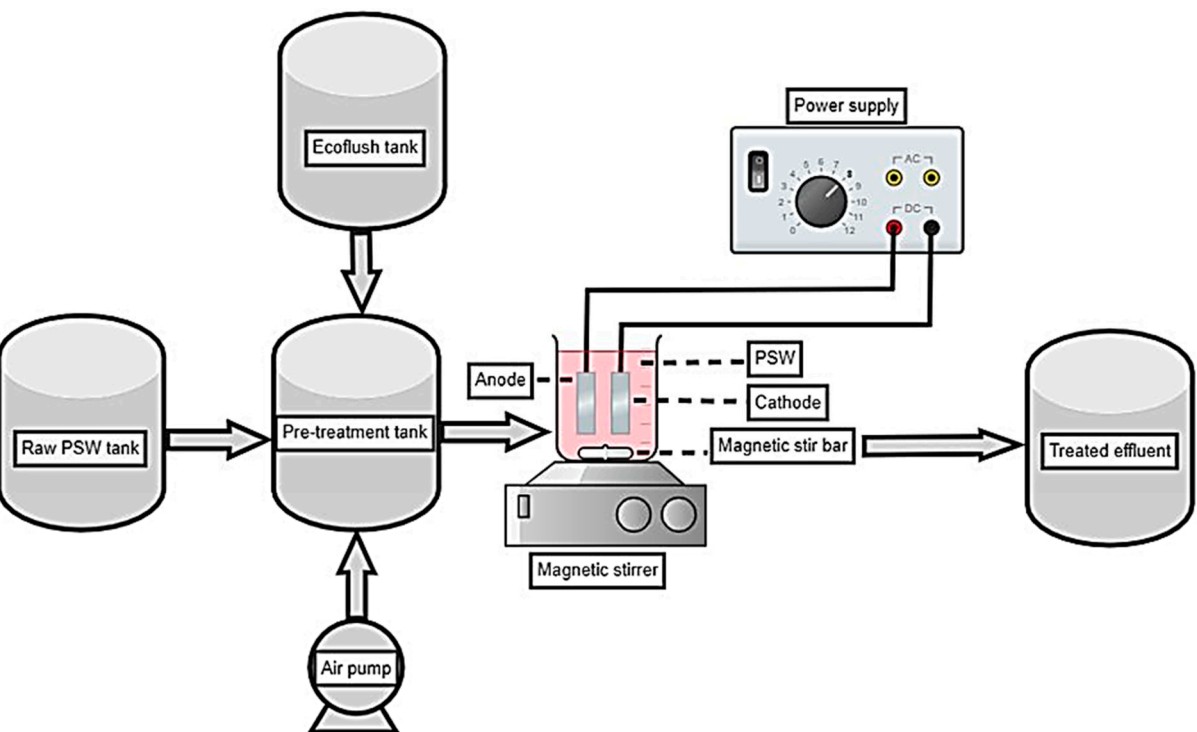

**Figure 1.** Schematic diagram of an integrated biological pre-treatment and electrocoagulation process used in this study.

### 2.3.2. Electrocoagulation Experiments

Electrocoagulation treatment was carried out in a plexiglass batch reactor having a dimension of 0.15 m × 0.10 m × 0.12 m with a working volume of 2 L. Iron was used as anode and cathode in a monopolar configuration with a total effective surface area of 807 cm$^2$. The electrodes were connected to a direct-current digital power supply (PS 8000 T, EA Elektro-Automatik, North Rhine Westphalia, Germany), characterised by the ranges 0–20 A for current and 0–16 V for voltage. The electrodes were fully submerged into PSW in the reactor and operated at a steady room temperature of 25 ± 0.5 °C during all experiments. The EC unit was constantly agitated at a rotational speed of 300 rpm by a magnetic stirrer (MS-H-Pro Plus, DLAB Instruments Ltd., Beijing, China). The optimisation of three numeric factors (initial pH (3–10), current density (13–72 A/m$^2$), reaction time (6–74 min)), and one categorical factor (Eco-flush™ (with or without)) was studied on the maximisation of two response variables: %COD reduction and %FOG reduction, using response surface methodology (RSM) based on a full factorial central composite design (CCD) with three levels for each factor (Table 2). The experimental design matrix was generated using Design-Expert® Software Version 12 (Stat-Ease, Inc., Minneapolis, MN, USA).

The initial pH of PSW was adjusted to the required value using 0.1 M hydrochloric acid (HCl) and sodium hydroxide (NaOH). Before each experimental run, electrodes were mechanically polished with abrasive paper and thoroughly rinsed with deionised water to remove any solid residue on the surface. Any stubborn impurities remaining on the surface were removed by dipping the electrodes for 5 min in 0.1 M HCl. At the end of each experimental run, the treated effluent was allowed to settle for 30 min. A sample of the supernatant was sampled and characterised for COD and FOG to ascertain the process efficiency. All the experiments were run as per the design matrix.

**Table 2.** Experimental design matrix showing variables and their respective outcomes.

| Run | Factor | | | | Response | |
|---|---|---|---|---|---|---|
| | A: pH | B: Current Density (A/m²) | C: Reaction Time (Mins) | D: Ecoflush™ | COD Reduction (%) | FOG Reduction (%) |
| 1 | 9 | 60 | 60 | With | 16.1 | 99.7 |
| 2 | 5 | 60 | 20 | With | 12.1 | 99.3 |
| 3 | 5 | 25 | 60 | Without | 76.7 | 99.9 |
| 4 | 7 | 43 | 40 | Without | 65.9 | 99.9 |
| 5 | 5 | 60 | 60 | With | 20 | 99.5 |
| 6 | 9 | 25 | 20 | With | 29 | 95.9 |
| 7 | 7 | 43 | 40 | Without | 62.4 | 99.9 |
| 8 | 5 | 60 | 60 | Without | 85.7 | 99.9 |
| 9 | 7 | 72 | 40 | With | 20.8 | 99.7 |
| 10 | 7 | 43 | 40 | With | 23 | 99.7 |
| 11 | 10 | 43 | 40 | Without | 63.7 | 98.6 |
| 12 | 5 | 25 | 20 | Without | 69.9 | 99.5 |
| 13 | 7 | 43 | 40 | With | 25.4 | 99.8 |
| 14 | 7 | 43 | 40 | Without | 64 | 99.9 |
| 15 | 7 | 43 | 40 | With | 20.3 | 99.9 |
| 16 | 7 | 13 | 40 | With | 22.5 | 93.7 |
| 17 | 9 | 60 | 20 | With | 29.1 | 99.9 |
| 18 | 7 | 43 | 6 | Without | 56.8 | 99.7 |
| 19 | 7 | 72 | 40 | Without | 70.4 | 99.9 |
| 20 | 7 | 43 | 40 | Without | 66.1 | 99.9 |
| 21 | 7 | 43 | 74 | With | 23 | 99.9 |
| 22 | 9 | 60 | 60 | Without | 68.8 | 99.9 |
| 23 | 7 | 43 | 40 | With | 22.7 | 99.5 |
| 24 | 9 | 60 | 20 | Without | 61.3 | 99.8 |
| 25 | 5 | 25 | 60 | With | 11.3 | 99.9 |
| 26 | 9 | 25 | 60 | Without | 62.7 | 99 |
| 27 | 7 | 13 | 40 | Without | 63.2 | 99.3 |
| 28 | 9 | 25 | 60 | With | 54.2 | 95.9 |
| 29 | 7 | 43 | 40 | With | 13 | 99.5 |
| 30 | 7 | 43 | 40 | Without | 65.9 | 99.9 |
| 31 | 10 | 43 | 40 | With | 31 | 98.3 |
| 32 | 7 | 43 | 40 | Without | 65.6 | 99.9 |
| 33 | 5 | 60 | 20 | Without | 80.3 | 99.9 |
| 34 | 7 | 43 | 6 | With | 13.5 | 98.1 |
| 35 | 7 | 43 | 74 | Without | 75.6 | 99.9 |
| 36 | 3 | 43 | 40 | With | 4.7 | 99.9 |
| 37 | 7 | 43 | 40 | With | 9.1 | 99.9 |
| 38 | 5 | 25 | 20 | With | 2 | 99.8 |
| 39 | 9 | 25 | 20 | Without | 63.7 | 99.1 |
| 40 | 3 | 43 | 40 | Without | 66.7 | 99.9 |

### 2.4. Modelling and Statistical Analysis

Design-Expert® Software allowed the fitting of quadratic empirical models onto the experimental data using multiple regression analysis. The software performed multiple regression analysis, including ANOVA at 95% confidence interval (CI), to evaluate the interactions between process variables and the responses. The relationship between the responses and four independent variables were evaluated by developing the second-order polynomial mathematical models (Equation (6)). The matrix, experimental range, and respective responses are presented in Table 2. The fitted models only included significant terms ($p > 0.05$) except when maintenance of the hierarchal structure was required. Optimisation analysis was performed to find combinations of process variables that would maximize %COD and %FOG reduction using the proposed best-fitting model equations.

$$Y = \beta_o + \sum_{j=1}^{k} \beta_j X_j + \sum_{j=1}^{k} \beta_{jj} X_j^2 + \sum_i \sum_{<j=2}^{k} \beta_{ij} X_i X_j + e_i \tag{6}$$

where $Y$ is the response; $\beta_o$ is the model intercept coefficient; $\beta_j$, $\beta_{jj}$, and $\beta_{ij}$ are the interaction coefficients of linear, quadratic, and second-order terms, respectively; $X_i X_j$ are independent variables ($i$ and $j$ range from 1 to $k$); $k$ is the number of independent parameters ($k = 4$ in this study), and $e_i$ is the error. The theoretical optimal conditions were repeated to practically confirm the results.

## 3. Results and Discussion

### 3.1. Characteristics of Poultry Slaughterhouse Wastewater

Comprehensive PSW analysis results (Table 3) were compared to those reported by previous studies [7,16]. Interestingly, wastewater from the same type of slaughterhouse industry exhibited a wide variation in the characteristics. This substantial variation indicates that the characteristics of PSW are site-specific and largely depend on local conditions as a result of different operational requirements and techniques [17]. The PSW effluent had high electrical conductivity, which is essential for the optimum performance of EC. It eliminates the need to add a supporting electrolyte required to facilitate current passage during treatment, thereby reducing electrical energy consumption and operating costs [18]. The maximum concentrations of TSS (8319 mg/L) were eight times higher than the stipulated limit value for discharge into the municipal sewer [19]. Compared with the effluent discharge standards of the City of Cape Town [19], PSW presented pH at the acidic region (6.19–7.24) and was the only parameter that did not exceed the limit.

**Table 3.** Characteristics of poultry slaughterhouse wastewater.

| Parameters | Units | Permissable Levels [a] | [16] | [7] | Raw PSW | Biological Pre-Treatment | Treated Effluent |
|---|---|---|---|---|---|---|---|
| | | Range | Average | Range | Range | Range | Range |
| pH | – | 12 | 7.95 | 7.03–8.23 | 6.19–7.24 | 5.48–6.75 | 3.43 |
| Electrical conductivity | mS/cm | 500 | 49.87 | 1.36–3.04 | 0.95–2.47 | 2.97–3.45 | 4.97 |
| TDS | Ppm | 4000 | – | – | 634–1701 | 953–1620 | $2.5 \times 10^6$ |
| Salinity | Ppm | – | – | – | 489–1395 | 961–1328 | $2.67 \times 10^6$ |
| Turbidity | NTU | – | – | – | 316–>1000 | 278–887 | 1.78 |
| COD | mg/L | 5000 | 3810 | 3968–5239 | 3750–14681 | 798–6490 | 1770 |
| SS | mg/L | 1000 | 46.45 | 475–1800 | 405–8319 | 106–247 | 234 |
| FOG | mg/L | 400 | – | 50–407 | 280–1668 | <1.0–183 | 2.5 |
| $NH_4^-N$ | mg/L | – | – | 20–38 | 53–312 | 92–219 | 120 |
| $PO_4^{-3}$ | mg/L | 25 | – | 72.25–190.48 | 30.8–56 | 134–178 | 9.4 |
| Heterotrophic Plate Count | cfu/mL | – | – | – | >3000 | – | 1 |
| Total coliforms | cfu/100 mL | – | 110,000 | – | >2000 | – | Not detected |
| *E. coli* | cfu/100 mL | – | – | – | >2000 | – | Not detected |

[a] Maximum limit of permitted discharges of wastewater and industrial effluent [19]. – Not indicated.

As expected, the COD and FOG concentrations were very high, which may be related to the nutrition and size of the birds slaughtered at the time of sampling [20]. There was a high ammonia nitrogen content, and PSW is expected to contain high concentrations of nitrogen in the protein from the blood [17]. The PSW also had a foul odour, that of "rotten eggs," which may be due to the presence of compounds in the digestive tracts of animals, such as proteins, fats, and carbohydrates which undergo microbial decomposition under aerobic conditions and release sulphides into the aqueous environment [21]. The primary odour source in PSW is hydrogen sulphide gas, volatile organic compounds such as volatile fatty acids, nitrogenous compounds, and organic particulate material. The high concentrations of organic matter in the form of FOG present a challenge associated with the treatment of PSW [22]. Hence, a pre-treatment unit may be required [12,13].

### 3.2. Evaluation of the Biological Pre-Treatment Performance

During the biological pre-treatment, 85% to 99% FOG reduction was achieved. Hydrocarbon chains in the FOG and other organic matter were weakened by hydrolytic enzymes present in the Ecoflush™ [12]. One of the most significant aspects was the low readily biodegradable COD (20% to 50%), which was even lower than previously reported in the literature. The post-biological effluent exhibited lower turbidity and odour, but a higher electrical conductivity was observed due to increased $NH_4^+$. The hydrolysis of proteins produces amino acids and ammonia while the bubbling of air in the Eco-flush™ reactor promoted the oxidation of hydrogen sulfide, resulting in odour reduction. Similarly, Del Pozo and Diez [23] found that the organic matter removal efficiency was very high (93%), while the nitrogen removal efficiency was initially low (29%) but later increased (up to 82%) as the aerated volume was increased. This would be due to the increased aerobic solid retention time (SRT) to encourage the growth of autotrophic nitrifying organisms (ANOs) that are known to be obligate aerobes and slow growers, and hence easily washed out of the system with lower aerobic retention times. It is conceivable that the raw PSW may have contained some aerobic bacteria that was present in the balancing tank at the time of sampling. These were then stimulated during the aeration of the Eco-flush™ reactor. Kibangou et al [24] observed the presence of bacteria and methanogenic archaea in raw tannery wastewater sampled from a balancing tank. Nonetheless, a good combination of anaerobic, anoxic, and aerobic processes is required for the biological removal of nutrients (N and P) [23]. This would be to cater for denitrification (anoxic zones to accommodate facultative heterotrophs that mediate the process) and P removal (phosphorus-accumulating organisms would ideally require alternating anaerobic and aerobic periods to carry out the metabolisms). A positive side to including the anoxic zone is (i) a lower effluent nitrate concentration and (ii) a reduction in the oxygen requirement for organic breakdown, due to some of the organic material being used for the process of denitrification.

Although the biological treatment was able to provide high organic-removal efficiencies in terms of FOG, a subsequent treatment process, such as EC treatment, is a viable option for handling a wide variety of other high levels of organics such as BOD and COD that could improve the final PSW quality characteristics. The slaughterhouse, as previously stated, produces high-strength wastewater with high COD which requires extended aeration when treated aerobically. Direct application of an aerobic treatment unit is associated with high costs of aeration and sludge disposal, necessitating an anaerobic pre-treatment stage [3]. For this reason, anaerobic pre-treatment is an efficient and cost-effective solution for this kind of wastewater as it significantly reduces sludge volume and beneficially produces biogas [17].

### 3.3. Mono and Synergistic Effect of Operational Factors on Process Efficiency

The results from this study indicated the significant impact ($p < 0.05$) of pH, retention time, and the interaction of pH and current density on both COD and FOG reduction (Table 4). However, Ecoflush™ was not significant ($p > 0.05$) in the removal of FOG due to the high efficiency (96–100%) of EC in removing FOG and hence its addition did not positively impact FOG removal. The integrated system achieved 93–100% FOG removal. Work by [12,13,25] reported high 89–100% FOG while treating PSW using Ecoflush. Although FOG is known to negatively impact the efficiency of EC due to its adherence to the surface of electrodes, causing their insulation, its impact was not observed in this study. This may have been due to the operating current density range used in this study.

**Table 4.** Analysis of variance for chemical oxygen demand (COD) and fats, oils, and grease (FOG) reduction.

| COD | | | | FOG | | | |
|---|---|---|---|---|---|---|---|
| **Source** | **Mean Square** | **F-Value** | ***p*-Value** | **Source** | **Mean Square** | **F-Value** | ***p*-Value** |
| Model | 2511.91 | 77.77 | <0.0001 | Model | 7.17 | 13.33 | <0.0001 |
| A-pH | 180.58 | 5.59 | 0.0250 | A-pH | 11.16 | 20.75 | <0.0001 |
| B-Current density | 60.71 | 1.88 | 0.1809 | B-Current density | 9.99 | 18.59 | 0.0002 |
| C-Reaction time | 340.44 | 10.54 | 0.0029 | C-Reaction time | 3.37 | 6.27 | 0.0184 |
| D-Ecoflush | 23874.69 | 739.13 | <0.0001 | D-Ecoflush | 1.93 | 3.58 | 0.0688 |
| AB | 325.17 | 10.07 | 0.0036 | AB | 11.13 | 20.70 | <0.0001 |
| AD | 1270.52 | 39.33 | <0.0001 | AD | 0.3043 | 0.5660 | 0.4581 |
| BC | 65.10 | 2.02 | 0.1664 | BD | 0.0379 | 0.0706 | 0.7924 |
| BD | 118.44 | 3.67 | 0.0654 | $A^2$ | 2.51 | 4.66 | 0.0395 |
| CD | 0.6212 | 0.0192 | 0.8907 | $B^2$ | 9.78 | 18.18 | 0.0002 |
| BCD | 138.55 | 4.29 | 0.0474 | ABD | 4.69 | 8.72 | 0.0063 |
| Residual | 32.30 | | | $B^2D$ | 2.75 | 5.11 | 0.0318 |

A better COD reduction was obtained when the EC unit worked at lower pH, higher current density, and retention time (RT). It is known that the amount of current density determines the coagulant dosage and size of the bubble production, and hence affects the growth of flocs [26]. However, current density had an insignificant ($p > 0.05$) impact on the integrated system for COD reduction and on the EC unit for FOG reduction. It had a significant ($p < 0.05$) impact on the integrated system during FOG removal. Lower FOG-removal efficiencies (93%) were observed at opposite extremes of the interaction between pH and current density due to the impact of these factors on Ecoflush and EC, respectively. The high %COD and %FOG reduction in acidic medium agrees with the findings of [2,27].

As shown in Figure 2A–D, the treatment process achieved up to 99% FOG reduction without Ecoflush™ while operating at initial pH between 2 and 8. This indicates that iron electrodes used in the present study operate best in acidic, neutral, and slightly alkaline pH, producing mostly $Fe^{2+}$ around pH 8 but generating $Fe^{3+}$ species as the pH lowers [28]. This was in agreement with a study by [29]. Operating pH influences the growth rates of microorganisms as well as the bioavailability of compounds that may stimulate or inhibit microorganisms. The majority of aerobic bacteria and even distribution of un-ionised and ionised pollutants such as $H_2S$ and $NH_3$ is at neutral pH [30]. After EC treatment, pH changes in the effluent were observed. This is attributed to hydrogen evolution and the generation of $OH^-$ ions at the cathodes [26]. In the present study, the initial pH (6.19–7.24) of raw PSW was ideal for both treatment systems and this eliminates the need for pH adjustment.

As expected, process efficiency linearly improved with an increase in RT. This is because, with an addition in the electrolysis time, more ions will be dissolved in the wastewater, leading to an increase in floc formation. It should be noted that the higher the reaction time, the more power is consumed [8]. However, the increases became insignificant above 20 min retention time. The interactions of RT with other factors on the process efficiency were insignificant ($p > 0.05$) except with Ecoflush™ for FOG removal. The literature indicates that the optimal reaction time for oily wastewater treatment ranges between 20–30 min [2,5,10].

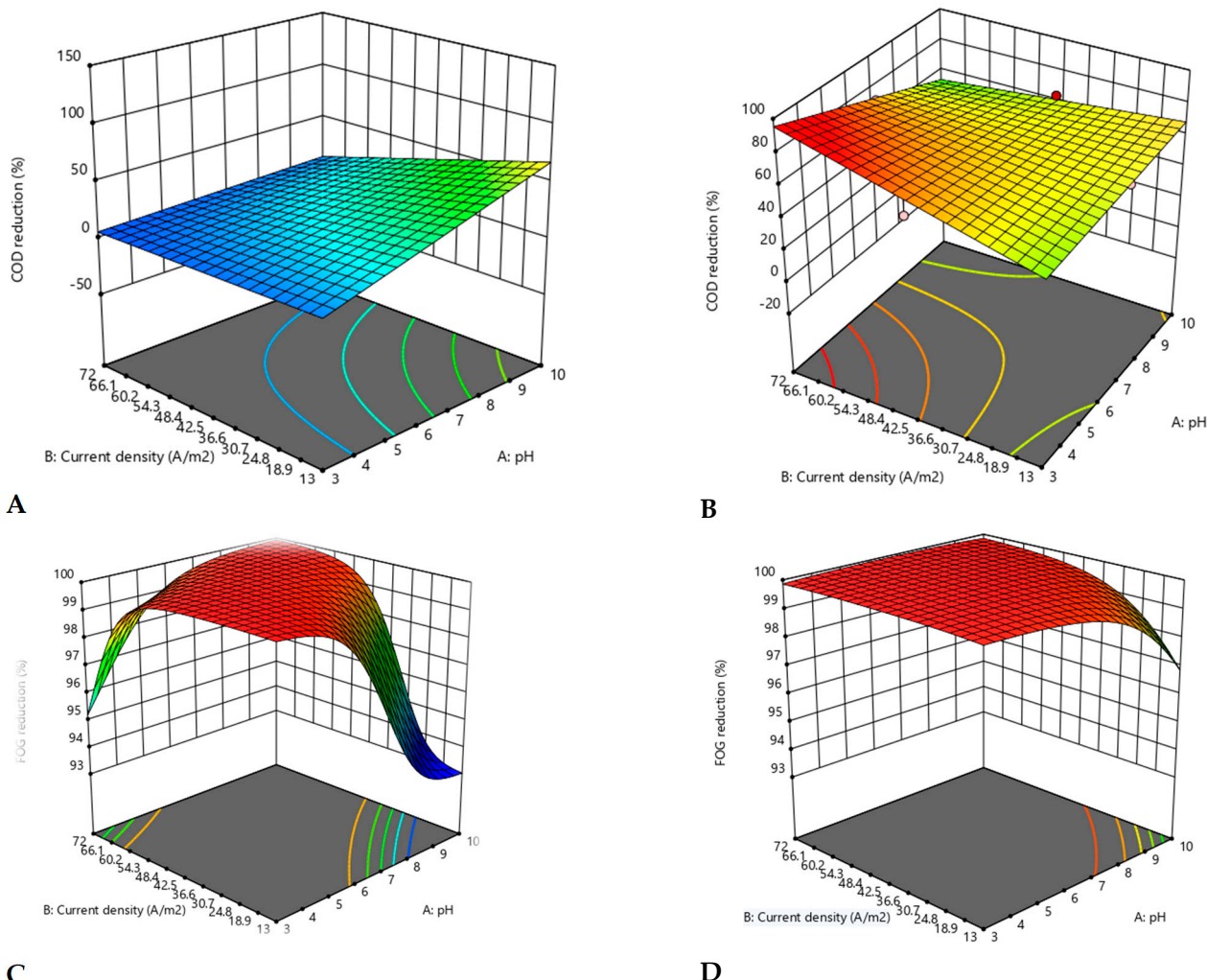

**Figure 2.** Effect of operating parameters on process efficiency: (**A**)—current density and pH on COD reduction (with Ecoflush™), (**B**)—current density and pH on COD reduction (without Ecoflush™), (**C**)—current density and pH on FOG reduction (with Ecoflush™); (**D**)—current density and pH on FOG reduction (without Ecoflush™).

*3.4. Process Optimisation and Analysis of Variance*

The fractional design space (FDS = 0.99) and the signal:noise ratios were sufficiently greater than the recommended 0.8, while the adequate precision for all the empirical models were desirable >4 (Stat-Ease, Inc., Minneapolis, MN, USA). The correlation coefficients ($R^2$) and adj. $R^2$ values which corrected the $R^2$ values in terms of sample size and several model terms indicated that only 4.8 and 3.6% for COD, and 16 and 22% for FOG could not be explained by the models, respectively.

The ANOVA results indicated that all models were significant (F test, $p < 0.05$), and there were minimal chances that this may have occurred due to noise (Table 5). The general quadratic polynomial equations did fit the %COD and %FOG reduction data very well (lack of fit: F test, $p \leq 0.05$)]. Therefore, Equations (7)–(10) were used to navigate the design space and to optimise the cumulative %COD and %FOG reduction as plotted in Figure 2. Based on the interest in maximising COD and FOG reduction, the theoretical optimum operating conditions at a desirability 1 were determined as pH = 3.1, current density = 66.9 A/m², retention time = 74 min, and without Ecoflush™. These optimum conditions were expected to achieve a reduction efficiency of 100% for both COD and FOG.

**Table 5.** Summary of the statistical results of the fitted models.

| Parameter | COD Model | FOG Model |
|---|---|---|
| $R^2$ | 0.9640 | 0.8396 |
| Adjusted $R^2$ | 0.9517 | 0.7767 |
| Predicted $R^2$ | 0.9109 | 0.6088 |
| Adequate precision | 27.9845 | 15.5485 |
| Mean | 43.95 | 3.06 |
| SD | 5.68 | 0.7333 |
| %CV | 12.93 | 23.96 |
| PRESS | 2320.41 | 36.72 |

CV = coefficient of variation; PRESS = predicted residual error sum of squares.

For COD reduction with Ecoflush™:

$$Y_1 = -76.94884 + 10.07236A + 1.37447B + 0.770394C 0.128781AB - 0.014168BC \tag{7}$$

For COD reduction without Ecoflush™:

$$Y_2 = 34.92281 + 3.39263A + 0.939701B + 0.070888C - 0.128781AB + 0.002644BC \tag{8}$$

For FOG reduction with Ecoflush™:

$$Y_3 = 6.38443 - 1.16837A + 0.028399B + 0.017488C + 0.039282AB - 0.067013A^2 - 0.002884B^2 \tag{9}$$

For FOG reduction without Ecoflush™:

$$Y_4 = 1.53005 + 0.249074A + 0.055609B + 0.017488C + 0.008364AB - 0.067013A^2 - 0.000893B^2 \tag{10}$$

*3.5. Validation of Process Optimum Conditions*

Additional experiments were conducted to validate the theoretical optimal conditions (pH = 3.05, current density = 66.9 A/m$^2$, reaction time = 74 min, and without Ecoflush™). According to the repeat experiments, 92.37% COD removal and 99.85% FOG reduction were achieved (Table 6). The actual experimental removal efficiency and the model prediction data were in very close agreement with less than 8.0% and 0.04% error for COD and FOG, respectively. This confirms the models' accuracy in predicting reduction efficiencies. These results are in agreement with [10,29] who achieved 90.4% and 85% COD reduction while using EC, respectively. The treated effluent also met the stipulated discharge standards as shown in Table 3. The closeness of the present study's results to that of the literature shows the efficiency of EC in treating PSW. However, it can be noted that the optimum electrolysis time in the present study is more than what is reported in the literature (Table 6). This could lead to high economic costs as EC requires a constant power supply produced from non-conventional energy sources. As a result, energy sources such as solar power, hydroelectric power, geothermal energy, and other renewable energy sources should be considered because they are more sustainable than fossil fuels [8]. Furthermore, RSM led to accurate modelling of COD and FOG removal.

**Table 6.** Summary of studies focusing on electrocoagulation treatment of poultry slaughterhouse wastewater.

| Evaluated Factors and Conditions | Electrode/Connection Type | Optimum | Efficiency (%) | References |
|---|---|---|---|---|
| Initial pH 6.7 | Al-Al | 2 | 93% COD. 90% FOG | [10] |
| Current density (25–200 A/m$^2$) | Fe-Fe | 150 A/m$^2$ | 85% COD. 98% FOG | |
| Electrolysis time (2.5–40 min) | Monopolar | 25 min | | |
| Initial pH (3–7) | Al-Al | 3 | 85% COD | [2] |
| Current density (0.5–2 mA/cm$^2$) | Monopolar | 1.0 mA/cm$^2$ | | |
| Electrolysis time (5–60 min) | | 30 min | | |
| Stirring speed (100–250 rpm) | | 150 rpm | | |
| Initial pH (3–9) | Fe-Fe | 3 | 95.5% COD | [27] |
| Current density (30–50 mA/cm$^2$) | Monopolar | 50 mA/cm$^2$ | | |
| Electrolysis time (15–90 min) | | | | |
| Supporting electrolyte (0.05–0.1 mg/L) | | | | |
| Initial pH (7.8) | Al-Al | | 94.4% COD | [31] |
| Current density (10–25 mA/cm$^2$) | Fe-Fe | | 81.1% COD | |
| Electrolysis time (60 min) | Monopolar | | | |
| Supporting electrolyte (0.05–0.1 mg/L) | | 0.05 mg/l | | |
| Stirring speed (100 rpm) | | | | |
| Initial pH (6.11–6.50) | Mild steel or Al | Mild steel | 82% COD | [32] |
| Electrolysis time (10–90 min) | | Through 60 or 90 min | 99% FOG | |
| Current intensities (1.0–3.0 A) | Monopolar | | | |
| Current intensities (0.3–1.5 A) | Bipolar | Bipolar | | |
| Initial pH (6.5) | Al-Al | | 95.6% COD. 92.5% FOG | [29] |
| Current density | Fe-Fe | 0.014 A/cm$^2$ | 94.5% COD. 95.3% FOG | |
| Electrolysis time (2.5–40 min) | Monopolar | 25 min | | |
| Stirring speed (300 rpm) | | | | |
| Initial pH (2–8) | Al-Al | 3 | 85% COD | [33] |
| Current density (1 mA/cm$^2$) | Monopolar | 1 mA/cm$^2$ | | |
| Electrolysis time | | 20 min | | |
| Current density (30 A/cm$^2$) | Al-Al | | 86% COD | [34] |
| Electrolysis time (40 min) | | | | |
| Initial pH (4) | | 7.5 | | |
| Initial pH(6.4) | Al-Gr (graphite) | | 76–85% COD | [35] |
| Current density (3–15 mA/cm$^2$) | | 3 A/cm$^2$ | 93–99% colour | |
| Electrolysis time (3–75 min) | | | 95–99% TSS | |
| Initial pH (3–10) | Fe-Fe | 3.05 | 92.37% COD | Present study |
| Electrolysis time (6–74 min) | Monopolar | 74 min | 99.85% FOG | |
| Current density (13–72 A/m$^2$). | | 66.9 A/m$^2$ | | |
| With/without Ecoflush$^{TM}$ | | Without | | |

## 4. Conclusions

The best operating conditions were obtained at pH of 3.05, a current density of 66.9 A/m$^2$, 74-min treatment time, and without Ecoflush™ using the RSM. These optimum EC process conditions produced a high-quality clarified effluent, 92.4% COD reduction, and 99% FOG reduction. The biological pre-treatment of PSW using Ecoflush™ resulted in 85–99% FOG reduction, 20–50% COD reduction, and an odourless effluent. However, the combination of both processes did not significantly improve the treatment efficiency when compared to the separate processes. This study showed that EC is a promising treatment method for PSW and the anticipated lipid inhibition did not lead to its ineffectiveness in treating PSW. Despite the low removal percentages of nitrogen, the study proved the feasibility of using Ecoflush™ for the removal of FOG prior to processes prone to lipid inhibition, such as anaerobic digestion.

**Author Contributions:** Writing—original draft preparation, P.V.N.; Supervision, M.B.; Funding acquisition, supervision M.B.; Writing—review and editing, P.V.N. and M.B.; Reviewing—L.G., T.H. and D.I. All authors have read and agreed to the published version of the manuscript.

**Funding:** This research was funded by National Research Foundation, Thuthuka Funding, Grant (No.138173).

**Institutional Review Board Statement:** Not applicable.

**Informed Consent Statement:** Not applicable.

**Data Availability Statement:** Not applicable.

**Acknowledgments:** The authors wish to acknowledge the National Research Foundation Thuthuka Funding, R017, for their financial contribution to this work.

**Conflicts of Interest:** The authors declare no conflict of interest.

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
