# Peer review of "Poultry Slaughterhouse Wastewater Treatment Using an Integrated Biological and Electrocoagulation Treatment System: Process Optimisation Using Response Surface Methodology"

_sustainability, doi:10.3390/su14159561_

Round 1
Reviewer 1 Report
This paper present a new method ofof an integrated biological and electrocoagulation treatment system in removing COD and FOG, it is interesting and significant. The manuscript is well written.
1. Compare with other treatment methods and list detailed advantages and disadvantages.
2. The conclusion should be concise and list important content only.
3. table 2: with, without cab be replaced by "√“and ”×”
4. line 266: but a higher electrical conductivity was observed due to increased NH4+ and most likely NO3−concentrations. This sentence is confusing, please rewrite.
5. Fig 2: Icons can be larger and clearer.
Author Response
The authours would like to thank the reviewers for reviewing the work and their invaluable comments has improved the paper.

Reviewer 2 Report
This is an interesting manuscript. However there are some problems that should be corrected
1) there is a need for English and general context language correction because there are some mistakes found eg see some of the following
you state
Alternative wastewater treatment strategies alternative to what?
you state
The 39 latter has grown in popularity
this is a strange stand alone sentence-at least give some reference for this statement
penalties imposed by municipal 34 councils on industries for the dis charge of untreated wastewater
why is dis charge two words? also this is so specific-it is not that only municipalities impose fines! it can be other offices or ministries, please correct the meaning of this sentence
you state
The efficient treatment of lipid-rich wastewater, such as PSW, poses considerable 61 technical and economic barriers due to its susceptibility to lipids that account for more 62 than 67% of the wastewater's particulate chemical oxygen demand (COD)
this is not an understandable sentence what you mean by susceptability? susceptability to what? susceptability of whom?
eg replace Permissive levels
with
Permissable levels
there are many more problems of comprehension please read and correct throughout. It is also important to use shorter more consice sentences
2) in lines 68-69: I dont understand why heavy metals are present in this kind of debris
3) lines 71-84: I think this is better for the discussion rather than the introduction
4) I am still a bit confused one what you tried here and why: an eco-flush design together with EC? you should summarize better and show the advantages and the design of the systems you propose and also the novelty of their combination (?) in the introduction
5) please do not use abbreviations that you have not explained-have you explained what FOG is?
6) you also say that it is the first time that RS methodology has been used but you do not say anything what that is and why it is important to be used here
7) you state such as stunning and slaughtering I am not sure what stunning means
8) I am a bit confused with all the debris that you present many of which is solid and not liquid-how this produces an influent? are there any filters of some size that withhold the solid waste? how homogeneous is this influent since there are so many different parts in it?Please elaborate how the liquid part of this debris was achieved and what do you describe as infuent wastewater here
9) you should quote all the method shown or the book of methods that contain them in the references section
10) I am not quite sure I understand what ecoflush was-is it a commercially available activated sludge? please explain a bit better. can it be used also at scaling up or only in the laboratory experiment? (in the discussion this answer(
11) for all equipment used please give name of equipment, name of company, city and country of origin. for all the methods used give a reference if there is one available in bibliography
12) in 2.5. Economic analysis of electrocoagulation I dont understand why you only focus on the cost of EC and not on the whole of the apparatus (Ecoflush+ EC). Also most of the information give here should be moved to the discussion here just give the method you used to calculate the costs
13) you have a different results and discussion section however you comment a lot on other studies in the results section! either merge results and discussion in one section or in the results show only the results and give all comments in the discussion only. In table 3 it should also have average +/- standard deviation
14) In relation to the previous comment: you describe a lot of important findings eg the interactions between parameters in the discussion section whereas clearly these are results! please show all your important results AND tables and figures in the results section where you should comment on the QUALITITAVE meaning of these results and everything else you compare with should be moved to the discussion setion. If you have not described the method in the methods section you cannot show results of this method
15) you state
Additional experiments were conducted to validate the theoretical optimal condi- 399 tions (pH=3.05, current density=66.9 A/m2 , reaction time= 74 minutes and without Eco- 400 flush™). you cannot do that you have to describe all the experiments in the methods otherwise you cannot show the results here. you should also decsribe the statistics that you did that you show the results here
16) I am not sure what Table 4 and Fig 2 as well as table 5 and Fig 3 overlap on. I believe only one of the two is needed everytime? please also explain what we see at least for table 4 fig 2 and fig 3 what we see, in the text
17) Table 7. Comparison of the treated effluent with the City of Cape Town's discharge standards. why is this necessary? you already shown the permissable standards in another table before?
18) as such all the relevant comparisons with other published work, that you have placed in introduction, methods and results should be transfered to the discussion and it should be compared to your findings. as such the references of the discussion are satisfactory in number. Please quote also the following ones
Marmanis, D.; Emmanouil, C.; Fantidis, J.G.; Thysiadou, A.; Marmani, K. Description of a Fe/Al Electrocoagulation Method Powered by a Photovoltaic System, for the (Pre-)Treatment of Municipal Wastewater of a Small Community in Northern Greece. Sustainability 2022, 14, 4323.
Chen, E.G.; Chen, X.; Yue, P.L. Electrocoagulation and Electroflotation of Restaurant Wastewater. J. Environ. Eng. 2000, 126,
858–863.
Khanitchaidecha, W.; Ratananikom, K.; Yangklang, B.; Intanoo, S.; Sing–Aed, K.; Nakaruk, A. Application of Electrocoagulation
in Street FoodWastewater. Water 2022, 14, 655.
Author Response

(The authors gave the same response as above.)

Reviewer 3 Report
The present study entitled “Poultry slaughterhouse wastewater treatment using an integrated biological and electrocoagulation treatment system: Process optimisation using response surface methodology” is an interesting paper. However there is lack of novelty in the research paper. A lot of similar work has been published already in different parts of the globe.
The manuscript may be revised in the light of the given comments.
Rewrite the abstract. Make if more result oriented.
There is no need to explain simple equation in the introduction. Pls add some previous similar work in the introduction part.
Biological Pre-Treatment needs more explanation
Economic cost of electro-coagulation is generally high as it needs constant power supply and high maintenance cost. Nowadays some low cost energy efficient and ecofriendly alternative techniques are also available. So author needs a comparative study for economic analysis. Try to ass a table to compare the economic analysis.
For the performance of the treatment process pls add results from the previous studies performed in different parts of the globe and compare them with the current study.
In the current study there is less stress on biological contaminants which are also big challenge in effluents in slaughterhouses.
Limitations of the study should also be the part of the study.
Conclusion part also needs major changes. Make it more informative and result oriented.
The references are very less. Add some recent appropriate references in the study.
Author Response

(The authors gave the same response as above.)

Round 2
Reviewer 2 Report
Ν/Α